# Characterization of RF System for MIR/THz Free Electron Lasers at Chiang Mai University

**Pitchayapak Kitisri [1], Jatuporn Saisut [1,2,3] and Sakhorn Rimjaem [1,2,3,*]**

[1] PBP-CMU Electron Linac Laboratory, Plasma and Beam Physics Research Facility, Department of Physics and Materials Science, Faculty of Science, Chiang Mai University, Chiang Mai 50200, Thailand

[2] Research Unit for Development and Utilization of Electron Linear Accelerator and Ultrafast Infrared/Terahertz Laser, Chiang Mai University, Chiang Mai 50200, Thailand

[3] Thailand Center of Excellence in Physics, Ministry of Higher Education, Science, Research and Innovation, Bangkok 10400, Thailand

[*] Correspondence: sakhorn.rimjaem@cmu.ac.th

**Abstract:** The establishment of the mid-infrared and terahertz free-electron laser (MIR/THz FEL) facility is ongoing at the PBP-CMU Electron Linac Laboratory (PCELL) in Chiang Mai University. The facility utilizes an S-band radio-frequency (RF) gun and a linear accelerator (linac) to generate and accelerate electron bunches. These electron bunches are accelerated in the RF gun and the linac using RF pulses with a frequency of 2856 MHz. Measuring the RF properties becomes essential, as the RF pulse information can be utilized to estimate the electron beam properties. To achieve the measurement results, we employed an RF measurement system comprising directional couplers, coaxial cables, attenuators, a crystal detector, and an oscilloscope. Prior to conducting measurements, the crystal detector and RF equipment were calibrated and characterized to ensure precise and reliable results. The electron beam energy estimation using the measured RF power was compared with the measured beam energies. The gun and the linac were operated with an absorbed RF power of 1.52 MW and an input power of 1.92 MW, respectively. The estimated electron beam energies were found to be 2.18 MeV and 15.0 MeV, respectively, closely aligning with the measured beam energies of 2.1 MeV and 14.0 MeV after the gun and linac acceleration. These consistent energy values support the reliability of our RF power measurement system and procedure.

**Keywords:** RF system; RF power; electron beam energy; RF electron gun; RF linac

## 1. Introduction

Infrared radiation is widely used in many applications, especially mid-infrared (MIR) radiation with a wavelength range of 3–30 micron and far-infrared (FIR) radiation including terahertz (THz) radiation with a wavelength of 0.03–3 mm. MIR radiation corresponds to the frequencies of covalent bonds in many molecules, especially biomolecules, with a non-destructive nature. Hence, it is widely used in bio- and medical science applications. As for FIR/THz radiation, it corresponds to molecular rotational-vibration frequencies. Additionally, it can penetrate non-metallic materials, reflect off metal surfaces, and be absorbed by water. Therefore, it finds applications in diverse areas such as chemistry, biology, agriculture, and medicine. To produce MIR and THz radiation with the required properties for specific applications, various techniques have been employed. One efficient technique is using a free-electron laser (FEL) capable of generating high-quality MIR and THz radiation. A notable feature of a FEL is its ability to produce short-pulsed radiation with high intensity and tunable wavelengths. Leveraging this advantage, along with the properties of MIR and THz radiation, it can be applied across various fields and applications. Consequently, MIR and THz FELs have found utility in numerous scientific research, industrial, and medical applications, including spectroscopy and imaging [1–4].

Currently, an electron accelerator system for an MIR and THz FEL light source has been designed, developed, installed and is being commissioned at the PBP-CMU Electron Linac Laboratory (PCELL) in Chiang Mai University [5]. A schematic drawing of the accelerator system is presented in Figure 1. The system consists of an electron source [6], a magnetic bunch compressor in a form of alpha magnet [7], a SLAC-type traveling-wave RF linac accelerator (linac) [8], steering magnets [9,10] and quadrupole magnets for electron beam transportation. Several beam diagnostic devices including current transformers [11], electron beam spectrometers [12,13], screen stations [14], and quadrupole scan set up [15] have been installed for measuring electron beam current, energy and energy spread, transverse beam size, and emittance, respectively. The electron source is a thermionic cathode radio-frequency (RF) gun, which has one and a half cell S-band standing-wave cavities with a side-coupling cavity. This RF gun can generate electron beams with a maximum energy within a range of about 2 to 2.5 MeV [6]. The energy and electron beam current can be adjusted within specific values by varying the RF power and the cathode filament temperature. After exiting the gun, the electron beam traverses through the magnetic bunch compressor in the form of an alpha magnet. Then, the electron beam is further accelerated in the linac to reach the expected energy in a range of about 10 to 25 MeV. This energy range is sufficient for generating THz/MIR FEL with properly optimized parameters of the beamline components [5,16,17]. Since the properties of electron beam depend strongly on the characteristics of the RF wave used to accelerate the electrons in the gun and the linac, proper optimization of the RF wave properties is crucial. In order to obtain precise information of the RF wave, reliable RF measuring devices and procedures are required. This work focused on establishing an RF measurement procedure with reliable precision for the measurement system. Energy measurement of the electron beam produced from the RF gun was performed using an alpha magnet with energy slits and a downstream current transformer (CT2) [12]. Furthermore, the beam after linac acceleration was measured with a dipole magnet (BD1) and a Faraday cup (FC1) located in the first beam dump, as shown in Figure 1. The measured results were used to estimate the RF power for comparison with the results from the RF measurements.

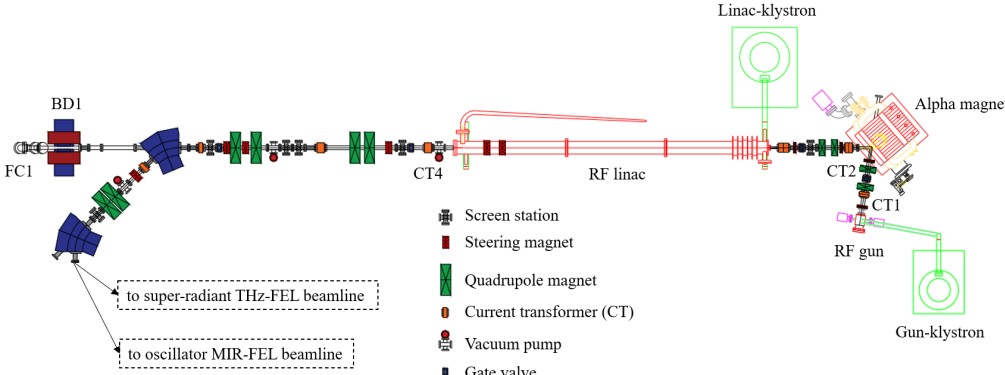

**Figure 1.** Schematic drawing of the accelerator system for the MIR/THz FEL light source at the PBP-CMU Electron Linac Laboratory (PCELL).

The RF power system is a crucial component of any RF accelerator, which plays a critical role in the overall operation and performance of the accelerator. It generates RF pulses with a specific frequency, pulse shape, and power level to accelerate the electron beam. At our facility, the RF wave is originally generated from an RF oscillator and propagates into a resonant cavity to form RF pulses with a frequency of 2856 MHz. After that, the RF wave is coupled into a pre-amplifier to obtain an power of several ten watts before being divided into two paths, for the gun and the linac. The RF power in each path is further increased using an amplifier and klystron. The final RF power level can be adjusted from 1 to 7 MW at a nominal RF pulse repetition rate of 10 Hz. The shape and width of the RF pulses are controlled by a pulse-forming network (PFN). The maximum pulse width of

the RF wave used for the RF gun is about 6 µs (FWHM), while it is about 8 µs (FWHM) for the linac. Figure 2 shows the schematic layout of the RF systems for the RF gun and the linac at PCELL.

Since the electron beam energy and pulse structure are directly correlated to the properties of the applied RF pulses, measurement of the RF power properties is required to ensure efficient beam operation. The energy gain of the electron beam produced from the RF gun can be determined from the RF power using the principle of the energy exchange between the RF field and the electrons accelerated in the RF gun. The relation of the stored energy from the RF field in the RF cavity $U$ and the power dissipated in the cavity wall per RF cycle $P_{cy}$ can be elucidated through the quality factor, given by $Q = \omega U / P_{cy}$, where $\omega$ represents the angular frequency. Consequently, the energy gain in electron per RF cycle is directly linked to the peak accelerating voltage $V_{RF}$ and the quality factor of the RF gun as

$$\Delta E_{gun}[MeV] = eV_{RF}Q, \tag{1}$$

where the accelerating voltage is related with the RF power $P_{RF}$, the specific shunt impedance $r_s$, and the effective length $d$ of the RF cavity as $V_{RF} = \sqrt{P_{RF}r_s d}$. The specific shunt impedance of the RF cavity can be approximated using a simplified form of the copper pill-box cavity that depends on the resonant frequency as $r_s(M\Omega/m) \approx 1.28\sqrt{f_{RF}(MHz)}$ [18]. For the linac acceleration, the energy gain $\Delta E_{linac}$ can be estimated from [8]

$$\Delta E_{linac}[MeV] = 10.48\sqrt{P_0[MW]} - 37.47 I_b[A], \tag{2}$$

where $P_0$ is the RF input power of the linac and $I_b$ represents the electron beam current. We applied the above two equations to estimate the electron beam energy after the gun and the linac acceleration by employing the measured RF power values. These estimate values were compared with the beam energy measurement results, as reported in Section 4.

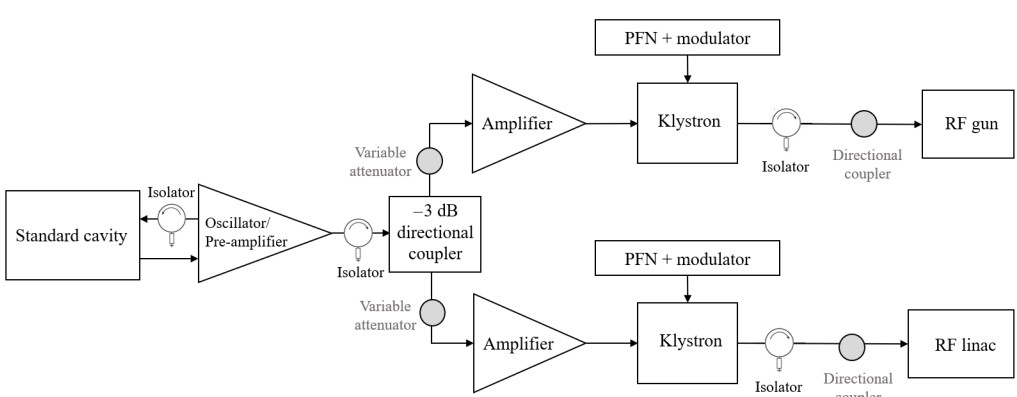

**Figure 2.** Schematic layout of the RF system for the RF gun and the linac.

## 2. Calibration and Characterization of RF Equipment

In this study, the RF measurement system comprised directional couplers, coaxial cables, RF attenuators, a crystal detector (Model 423B, Agilent (Hewlett-Packard), Agilent Technologies Inc., USA), and a digital oscilloscope (Model MSO5104, RIGOL Technologies, China). The RF power system, the RF gun and the linac are situated in the underground hall of our laboratory, while the crystal detector and oscilloscope are located in a separate control room. This physical separation ensures the radiation safety condition for operators and prevents radiation damage of the electronic equipment. The RF power $P_0$ feeding the RF gun or the linac can be calculated from

$$P_0 = P_d \times 10^{(a/10)}, \tag{3}$$

where $P_d$ is the input power of the crystal detector and $a$ is the absolute value of the total attenuation from the measurement system, which includes the attenuator of the directional coupler, the cables, and additional RF attenuators. For a reliable precision of RF measurement, a crystal detector calibration is required. Furthermore, the attenuator of the whole RF signal transport line must be known.

### 2.1. Crystal Detector Calibration

A crystal detector is a device that converts the RF power levels applied to the 50-$\Omega$ input connector, attached to a low-barrier Schottky diode, into proportional values of DC voltage [19,20]. The device possesses its own characterized calibration curve of the input RF power and the output DC voltage. Prior to utilizing this device, calibration of the curve is necessary. Moreover, there exists a possibility that the characterized curve may change over time. Hence, periodic calibration of the crystal detector is essential. In this work, there were two steps for calibrating the crystal detector. First, in order to accurately determine the input RF power at the end of the input wire, an RF power sensor (Model N8487A, Keysight Technologies, Inc., CA, USA) and a power meter (Model N1913A, Keysight Technologies, Inc., CA, USA) were employed, as illustrated in Figure 3. The input RF power was then recorded. Second, the RF power sensor and power meter were replaced with a crystal detector and a voltmeter. The DC voltage value was measured and recorded. These two steps were repeated for each variation in the RF input power. Figure 4 presents the calibration result for the crystal detector utilized for the RF power measurements presented in this paper. The result exhibited a quadratic polynomial relationship between the output DC voltage $V$ and the input RF power $P_d$ of the crystal detector, achieving a maximum goodness of curve fitting ($R^2 = 1$). The calibration equation for this crystal detector was determined as

$$P_d[mW] = 1.321 \times 10^{-5}V^2 + 1.121 \times 10^{-3}V[mV]. \tag{4}$$

This calibration process ensured the accuracy and reliability of the crystal detector used in our RF power measurements.

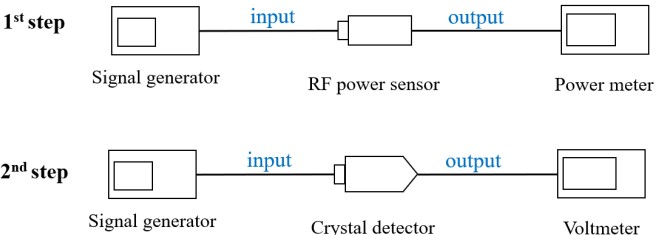

**Figure 3.** Schematic layout of the crystal detector calibration setups. The 1st step is to accurately determine the input and output RF powers. The 2nd step is to obtain the calibration data between the input RF power and the output DC voltage.

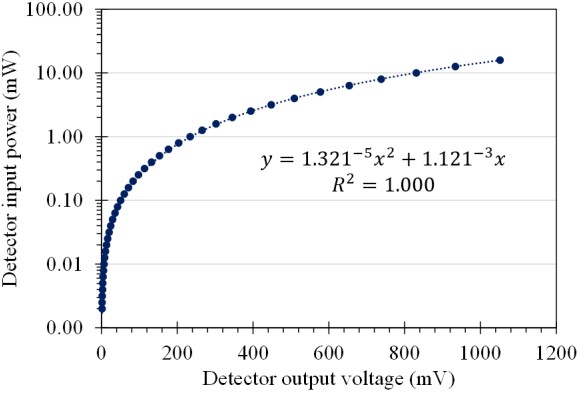

**Figure 4.** Crystal detector calibration curve with quadratic polynomial fitting equation.

## 2.2. Attenuation Characterization

To perform the RF power measurement, the attenuation value of the RF equipment used in the RF signal transport line is required. In this work, a vector network analyzer (Model R&S® ZVL6, Rohde & Schwarz GmbH & Co. KG, Germany) was used to characterize the attenuation value of the directional couplers, RF attenuators, and coaxial cables. Figure 5 presents an example setup of the directional coupler attenuation characterization. Table 1 shows the attenuation values of the RF devices that were used in the measurement system.

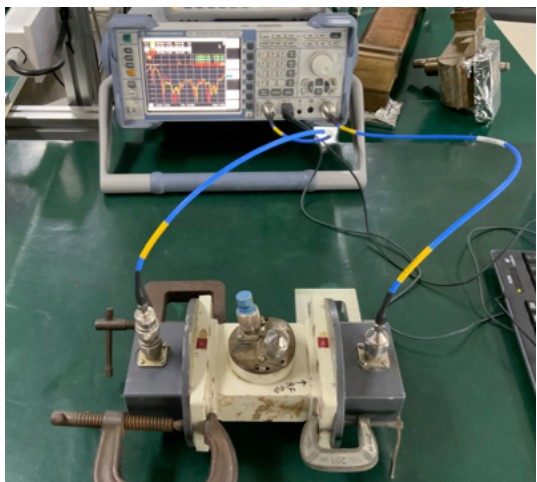

**Figure 5.** Attenuation characterization of the directional coupler for the RF gun.

**Table 1.** List of attenuation values of RF equipment.

| RF Equipment | Specification | Attenuation [dB] |
|---|---|---|
| Attenuation no. 2 | −10 dB | −9.994 |
| Attenuation no. 3 | −10 dB | −9.969 |
| Attenuation no. 4 | −3 dB | −2.862 |
| Attenuation no. 7 | −10 dB | −10.126 |
| Attenuation no. 9 | −5 dB | −4.821 |
| Attenuation no. 14 | −20 dB | −19.833 |
| Attenuation no. 17 | −19.4 dB | −19.510 |
| Attenuation no. 19 | −10 dB | −9.970 |
| Coaxial cable no. 1 | N/A | −23.323 |
| Coaxial cable no. 2 | N/A | −23.040 |
| Directional coupler no. 1 | N/A | −70.190 |
| Directional coupler no. 2 | N/A | −60.386 |

## 3. RF Power Measurements

After characterizing all RF equipment devices in the RF signal transport line, including calibration of the crystal detector, the input and output RF powers of both the gun's klystron and the linac's klystron were measured. This investigation aimed to assess the performance of both klystrons and determine the possible RF power levels they could provide.

### 3.1. RF Measurement of the Gun's Klystron

To measure the input RF power of the gun's klystron, the measurement system consisted of four RF attenuators (numbers 2, 3, 7, and 19), a crystal detector, and an oscilloscope. To measure the output RF power, the measurement system comprised the directional coupler number 1, coaxial cable number 1, attenuator number 3, a crystal detector, and an oscilloscope. During the measurements, the variable attenuator located upstream of the gun-amplifier (as shown in Figure 2) was adjusted in several steps. The oscilloscope captured the output voltages of the crystal detector, allowing for determination of the detector

input RF powers using Equation (4). Subsequently, the RF powers were calculated using Equation (3), considering the total attenuation of the measurement system, as listed in Table 1. The total attenuation of the measurement system for the input and output RF powers was −40.059 dB and −103.482 dB, respectively.

The relationship between the input and output RF powers of the gun's klystron with its power gain is illustrated in Figure 6. The results indicated that, as the input power increased, the output power increased, while the power gain decreased. The output RF power could reach a power level of 1.72 MW with a power gain of about 90,000 at an input power of about 19 W.

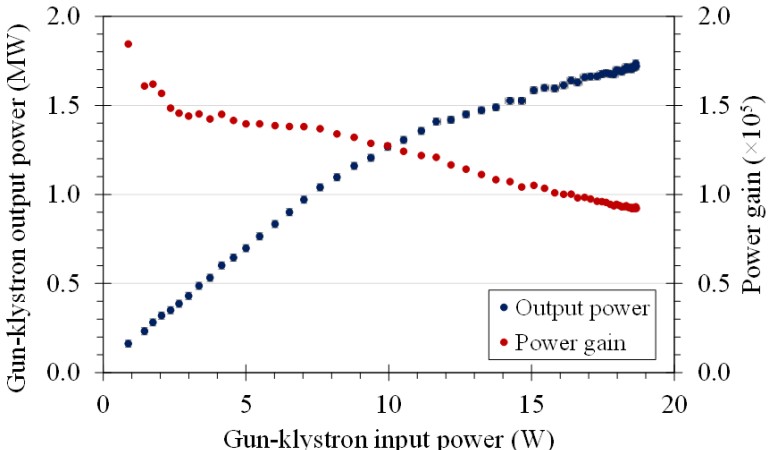

**Figure 6.** Relation between the input and output RF powers with power gain for the gun's klystron.

### 3.2. RF Measurement of the Linac's Klystron

The measurement system for the input RF power of the linac's klystron consisted of four RF attenuators (no. 2, 3, 14, and 17) along with a crystal detector and an oscilloscope. Similarly, the measurement system for the output RF power of the linac's klystron included the directional coupler no. 2, coaxial cable no. 2, 3 RF attenuators (no. 3, 4, 9), a crystal detector, and an oscilloscope. In both measurements, the variable attenuator for the input RF power was adjusted, and the oscilloscope monitored the voltage signal from the crystal detector. The crystal detector input powers were determined using Equation (4), and subsequently, the RF powers were calculated using Equation (3). The total attenuation values for the measurement systems of input and output RF powers were −59.306 dB and −101.078 dB, respectively.

The correlation between the input and output RF powers of the Linac's klystron with its power gain is presented in Figure 7. As the input power increased, the results demonstrated a concurrent increase in output power and a decrease in power gain. At the input power of about 78 W, the output power reached a power level of 1.92 MW with a power gain of about 2500. The output power started to saturate as the input power approached 50 W. It is noted that the current output RF power of the linac's klystron has not yet reached its maximum power level. Unfortunately, during the beam operation described in this paper, only two out of the five high-voltage steps could be utilized, due to a fault in the modulator transformer. Consequently, the achievable RF power was lower than the required value for generating the electron beam for the MIR-FEL. Further RF power characterization of the linac's klystron will be conducted after the replacement of the modulator transformer. According to the klystron's specifications, an output power in the range of 6–7 MW is expected. With this anticipated RF power, it should be feasible to obtain an electron beam energy higher than 22 MeV.

Additionally, we verified the reliability of our RF measurement system by utilizing an RF power sensor (Model Bird 5106D, Bird Technologies, USA) to measure the input power of the linac's klystron. The comparison of the measurement results between our RF

measurement system and the RF power sensor is illustrated in Figure 8. According to the results, error bars of the RF power measured from the RF power sensor in the lower power regime (0 to 20 W) are influenced by the fluctuation of the RF power signals, whereas error bars in the higher power regime (20 to 60 W) are caused by the noise level of the RF power sensor. However, the outcome clearly demonstrates a close agreement between the RF powers measured by our measurement system and those measured by the RF power sensor, further affirming the reliability of the RF measurement system at our laboratory.

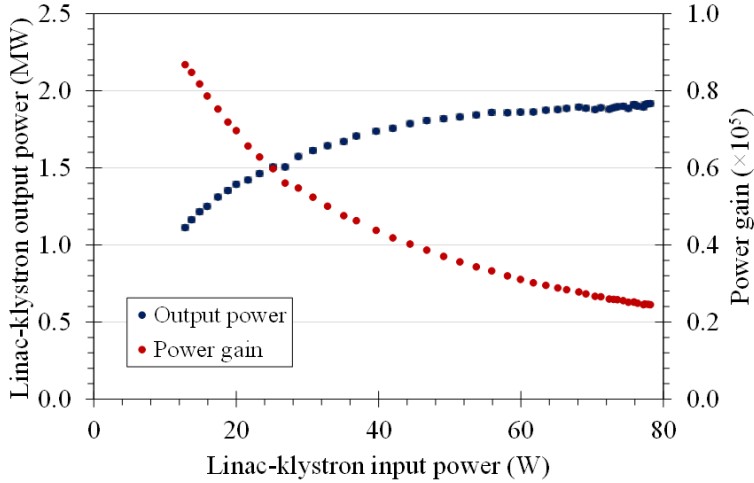

**Figure 7.** Relation between input and output RF powers with power gain for the linac's klystron.

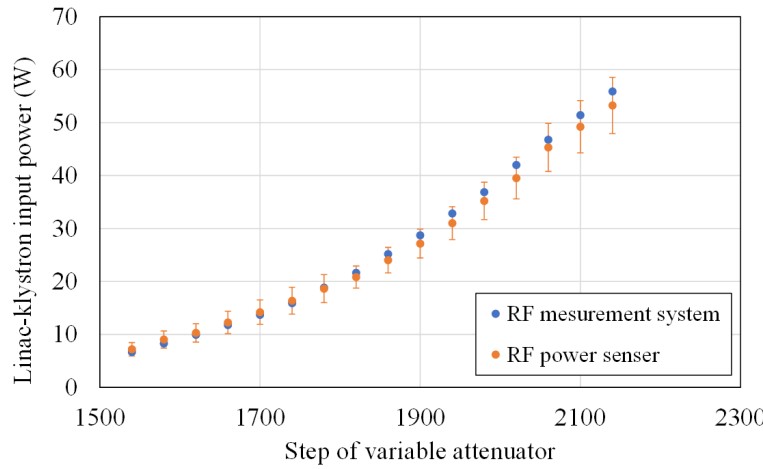

**Figure 8.** Comparison of the measurement results obtained from our RF measurement system and the RF power sensor for the linac's klystron input power.

## 4. Electron Beam Operation

To facilitate the electron beam commissioning process, we conducted measurements of the beam energy and current downstream of the RF gun and the linac. The RF waves were supplied to the gun and linac with input RF powers of 1.72 MW and 1.92 MW, respectively. The values mentioned here were the highest recorded during the measurement period reported in this paper, but they do not represent the maximum RF power capability of the klystron. According to the klystron's specifications, its maximum RF peak power is 7 MW.

To measure the electron beam energy downstream of the RF gun, an alpha magnet equipped with adjustable energy slits and a current transformer (CT2) was utilized, as depicted in Figure 1. The trajectory of an electron traveling within the magnetic field of the alpha magnet corresponded to both the magnetic field gradient and the energy of the

electron beam [21]. With an appropriate magnetic field gradient, electron beams could traverse the alpha magnet and reach the CT2. Consequently, the beam current signal was measured. During the measurement process, the energy slit positioned inside the alpha magnet's chamber was slowly adjusted until it cut the electron beam. As a result, a change in the current signal of the CT2 was observed, indicating the position of the electron beam within the alpha magnet. This procedure facilitated the determination of the energy of the electron beam. During beam operation, electron beams accelerated from the RF gun had energies of approximately 2.1 MeV. Moreover, the beam currents were carefully measured using a current transformer positioned downstream of the RF gun (CT1). The operation resulted in beam currents of approximately 750 mA.

To measure the energy of the electron beams accelerated by the RF linac, a dipole magnet and a Faraday cup were employed. The trajectory of an electron passing through the magnetic field of the dipole magnet corresponded to the magnetic field strength and the energy of electron [22]. When the bending angle was fixed, the appropriate magnetic field strength was adjusted to bend the electron to hit the Faraday cup downstream of the dipole magnet. Consequently, the signal from the Faraday cup was detected, allowing for the measurement of the electron's energy. During the beam operation reported in this paper, the electron beam energy was approximately 14.0 MeV after being accelerated through the linac. Furthermore, the beam currents were measured using a current transformer positioned downstream of the linac (CT4). The measurement result showed that the electron beams had currents of approximately 40 mA.

Based on the results of the RF power measurements discussed in Section 3, we utilized Equation (1) and its correlated relations to estimate the energy of the accelerated electron beam from the gun. The absorbed RF power of 1.52 MW, which was calculated from the difference between the input RF power (1.72 MW) and the reflected RF power (0.20 MW), was used to estimate the electron beam energy. The quality factor and the effective length of the RF gun were obtained from a previous work [6]. The effective lengths of the first and the second cavity of the RF gun were estimated based on the measured longitudinal electric field profile, resulting in 3.175 cm and 5.588 cm, respectively. The RF gun's quality factor was determined to be 12,979. Consequently, the energy gain of the electron beam per RF cycle accelerated by the RF gun was estimated to be 2.18 MeV. As for the linac, the accelerated electron beam energy was approximated using Equation (2). With an RF power of 1.92 MW and a beam current of approximately 40 mA, the measured energy measured was 13.0 MeV. Considering the initial energy gained from the RF gun, estimated to be around 2 MeV, the total energy of the electron was about 15.0 MeV after linac acceleration. These estimated values from the RF gun and the RF linac were consistent with the measured electron beam energies, which were 2.1 MeV and 14.0 MeV downstream of the RF gun and the RF linac, respectively. These results certainly demonstrated the reliability of our RF power measurement system.

## 5. Conclusions

At the PBP-CMU Electron Linac Laboratory, 2856-MHz RF waves are used to accelerate an electron beam in the gun and the linac with the aim of generating a MIR and THz FEL. To achieve the required electron beam energy, it was necessary to characterize the properties of the RF wave applied to both the gun and the linac. Before conducting the RF power measurements, calibration and characterization of the equipment used for measuring RF power, including the crystal detector, attenuators, coaxial cables, and directional couplers, were performed. In the case of measuring the powers of the gun's klystron, an input RF power of approximately 19 W resulted in a maximum output RF power of 1.72 MW, with a power gain of about 90,000. For the measurement of the power of the linac's klystron, only the first and second steps of the modulator high-power voltages could be utilized during this beam operation period. The output RF power reached a maximum value of 1.92 MW for an input RF power of 78 W with a power gain of about 2500 and the output power began to saturate as the input power approached 50 W.

In the electron beam operation, the gun and the linac were operated with input RF powers of 1.72 MW and 1.92 MW, respectively. The absorbed power in the gun was calculated to be 1.52 MW. Utilizing the information of the RF powers during the gun and linac operation, the electron beam energies were estimated to be 2.18 MeV and 15.0 MeV, respectively. During the beam operation, electron beam energy measurements were conducted, revealing beam energies of 2.1 MeV and 14.0 MeV after the gun and linac acceleration, respectively. These measured beam energies agreed well with the estimated values derived from the measured RF powers, confirming the reliability of the RF power measurement system and procedure. Based on the aforementioned results, the electron beam with an energy of 14.0 MeV successfully meets the beam energy requirement for the THz-FEL. However, to fulfill the MIR FEL's higher beam energy requirement of 22–25 MeV, an increase in the klystron's output power by adjusting the modulator high-power voltage will be implemented after replacement of the modulator transformer. According to the klystron's specifications, it can amplify the RF power up to about 7 MW. This value is sufficient to meet requirement of electron beam energy for the MIR FEL. The procedure and resulting data from the RF power measurements conducted in this work can serve as a reliable guideline for future RF power measurements and electron beam operations in our laboratory.

**Author Contributions:** Conceptualization: S.R., P.K. and J.S.; methodology: P.K., S.R. and J.S.; validation: S.R. and J.S.; formal analysis: P.K.; investigation: P.K.; resources: S.R. and J.S.; data curation: P.K.; writing—original draft preparation: P.K. and S.R.; writing—review and editing: S.R.; visualization: S.R.; supervision: S.R. and J.S.; project administration: S.R.; funding acquisition: S.R. All authors have read and agreed to the published version of the manuscript.

**Funding:** This research received funding support from Chiang Mai University and the NSRF via the Program Management Unit for Human Resources & Institutional Development, Research and Innovation (grant number B05F650022).

**Data Availability Statement:** The data that support the findings of this study are available from the corresponding author (S.R.) upon reasonable request.

**Acknowledgments:** This research received funding support from Chiang Mai University and the NSRF via the Program Management Unit for Human Resources & Institutional Development, Research and Innovation (grant number B05F650022). The authors extend their gratitude to the National Astronomical Research Institute of Thailand (NARIT) for supporting the devices used in the low-level RF measurements. Special thanks go to Dan Singwong for his valuable assistance and guidance in the calibration and characterization of the RF equipment.

**Conflicts of Interest:** The authors declare no conflicts of interest.

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
