# Peer review of "Characterization of RF System for MIR/THz Free Electron Lasers at Chiang Mai University"

_2571-712X, doi:10.3390/particles7020021_

Round 1

Reviewer 1 Report

Comments and Suggestions for Authors

Also, the manuscript (MS) is written in a good professional language, unfortunately I cannot suggest it for publication in its present form. Two main reasons turned me to this conclusion:

First, the MS presents a RF measuring system for evaluation, through some empiric calibration and relations, the energy of the accelerated electron bunches at the exit of RF gun and of the RF linac of MIR/THz Free Electron Lasers at Chiang Mai University. However, the method was experimentally checked only at a single value of the gun’s klystron RF power, and at a single value of the linac RF power. To my understanding, at least several measurements (RF power and the corresponding electron bunch energy) made at the same system's point for different levels of RF power, are requested for verification of the suggested method. Intension to apply the evaluation method in future for a much high bunch energy (22-25 MeV instead of 14-15 MeV in the current experiment) as mentioned in the conclusion, just strengthens a need in a more systematic verification of the method.

Second, the MS deals with a subsystem of the “MIR/THz Free Electron Lasers at Chiang Mai University”. However, the only citations to present the project and preliminary parts of the given research are theses and a B.Sc. independent study report which are unavailable at web. My very first WEB search reveals last year’s conference publication published at 13th Int. Particle Acc. Conf. – IPAC 2022 which is available at JACoW website. Why didn’t the authors cite this of similar sources (referred journal or conference publications)?

The following corrections are less crucial but also seem to be important:

·         p.1 line 34: “The measurements of electron energy were conducted…”. The method is explained much later, in section 4. Some short explanation seems to be in place also in the introduction.

·         p.3 line 79: “Figure 3 presents …”. Should not this be Figure 4 referred?

·         p.3 line 80-82: “The results indicate a slight degradation in the new calibration curve compared to the last two calibration curves, which can be attributed to the passage of time.” According to Figure 4, one can conclude that the measurements were made with the interval of about 10 years. If so, this indeed can explain some differences in the measured results. However, to my understanding this issue should be more clearly explained in the text. More of this, it seems that Eq. (4) takes estimation only of the resent calibration line (of three calibration lines given at Figure 4). The reasons should be explained, or this fact should at least be mentioned in the text.

Author Response

Response to Reviewer 1 Comments

Also, the manuscript (MS) is written in a good professional language, unfortunately I cannot suggest it for publication in its present form. Two main reasons turned me to this conclusion:

Point 1:

First, the MS presents a RF measuring system for evaluation, through some empiric calibration and relations, the energy of the accelerated electron bunches at the exit of RF gun and of the RF linac of MIR/THz Free Electron Lasers at Chiang Mai University. However, the method was experimentally checked only at a single value of the gun’s klystron RF power, and at a single value of the linac RF power. To my understanding, at least several measurements (RF power and the corresponding electron bunch energy) made at the same system's point for different levels of RF power, are requested for verification of the suggested method. Intension to apply the evaluation method in future for a much high bunch energy (22-25 MeV instead of 14-15 MeV in the current experiment) as mentioned in the conclusion, just strengthens a need in a more systematic verification of the method.

Response 1:

The reviewer's suggestion regarding performing multiple measurements of RF powers and corresponding electron bunch energies aligns perfectly with our intentions, and we do plan to carry out such comprehensive measurements in the near future. However, the primary focus of this manuscript is to present the methods and procedures used for characterizing the RF system at our facility. In this context, we conducted a demonstrative case with a single value of the gun RF power and a single value of the linac RF power. The purpose of this demonstration was to verify the effectiveness of our method and confirm its reliability. The current manuscript serves as an essential foundation for future investigations involving more extensive data collection and analysis. To clarify our intentions and facilitate the readers' understanding, more precise estimation formula and detailed explanation were added in this revised manuscript, specifically in lines 10-16, 58-69, 132-152, 165-176.

Point 2:

Second, the MS deals with a subsystem of the “MIR/THz Free Electron Lasers at Chiang Mai University”. However, the only citations to present the project and preliminary parts of the given research are theses and a B.Sc. independent study report which are unavailable at web. My very first WEB search reveals last year’s conference publication published at 13th Int. Particle Acc. Conf. – IPAC 2022 which is available at JACoW website. Why didn’t the authors cite this of similar sources (referred journal or conference publications)?

Response 2:

Thank you for bringing this to our attention. In this revised manuscript, we have updated the citations of the project and preliminary researches to refer to the IPAC2022 conference publications, which are now listed as references [1-3]. These references can be accessed on the website https://www.ipac22.org/.

Point 3:

The following corrections are less crucial but also seem to be important:

p.1 line 34: “The measurements of electron energy were conducted…”. The method is explained much later, in section 4. Some short explanation seems to be in place also in the introduction.

Response 3:

The methods for measuring the electron beam energy are explained in the Introduction section of this revised manuscript (lines 34-38).

Point 4:

p.3 line 79: “Figure 3 presents …”. Should not this be Figure 4 referred?

Response 4:

Thank you very much for pointing this out. We have changed the word from “Figure 3” to “Figure 4” on p.3 line 79 in the previous manuscript.

Point 5:

p.3 line 80-82: “The results indicate a slight degradation in the new calibration curve compared to the last two calibration curves, which can be attributed to the passage of time.” According to Figure 4, one can conclude that the measurements were made with the interval of about 10 years. If so, this indeed can explain some differences in the measured results. However, to my understanding this issue should be more clearly explained in the text. More of this, it seems that Eq. (4) takes estimation only of the resent calibration line (of three calibration lines given at Figure 4). The reasons should be explained, or this fact should at least be mentioned in the text.

Response 5:

After considering the reviewer's comment, we afraid that presenting both the old and new calibration results of the crystal detector could potentially confuse the readers. As we utilized only the new calibration result in the RF power measurement presented in this paper, we thus intend to present and discuss solely the new calibration result in this revised manuscript as written in lines 93-99 and in Figure 4.

Reviewer 2 Report

Comments and Suggestions for Authors

In this paper, the beam energy has been estimated from the measured RF power feeding the RF gun and accelerating structure and compared with the measured results. Although it seems important to derive the energy gain from the RF power input to the RF components, no new scientific findings have been presented.

 I consider the following revisions are necessary.

1.  It is impossible to determine the beam energy at the RF gun exit by measuring the RF power and using Equation 1. The assumption in line 137 of the text that all RF power contributes to the beam power is incorrect. An equation for the relation between RF power and beam energy using parameters of the RF gun cavities, such as Q-value, should be derived. Equation 1 should be corrected.

2.  Equation 3 is not necessary.

3.  Could "Figure 3" in text line 79 be a mistake for Figure 4?

4.  Fig. 6 and Fig. 7 should be shown with the horizontal axis as input RF power and the vertical axis as output RF power. The reader is not interested in the variable attenuator step. The text should also be revised.

5.  References should essentially  be to peer-reviewed journals. If the reference is to a PhD thesis or Master's thesis, in many cases the reader will not be able to find the reference. (If a DOI is given, it should be stated).

Comments on the Quality of English Language

Minor corrections are required.

Author Response

Response to Reviewer 2 Comments

In this paper, the beam energy has been estimated from the measured RF power feeding the RF gun and accelerating structure and compared with the measured results. Although it seems important to derive the energy gain from the RF power input to the RF components, no new scientific findings have been presented. I consider the following revisions are necessary.

Point 1:

It is impossible to determine the beam energy at the RF gun exit by measuring the RF power and using Equation 1. The assumption in line 137 of the text that all RF power contributes to the beam power is incorrect. An equation for the relation between RF power and beam energy using parameters of the RF gun cavities, such as Q-value, should be derived. Equation 1 should be corrected.

Response 1:

Thank you for the valuable guidance provided by the reviewer. In response to the feedback, we have replaced the formula in Equation 1 with a new one that presents the relation between RF power and beam energy using parameters of the RF gun cavities, which are quality value, shunt impedance, and resonant frequency. The detailed information about these parameters can be found in lines 58-69 in this revised manuscript. The updated results of the beam energy estimation using the new equation are now presented in lines 145-174.

Point 2:

Equation 3 is not necessary.

Response 2:

Thank you for the reviewer's opinion. However, we have decided to keep this equation in the manuscript to ensure that readers can easily follow the calculation of the RF powers that we have measured using our measuring system.

Point 3:

Could "Figure 3" in text line 79 be a mistake for Figure 4?

Response 3: 

Thank you very much for pointing this out. We have changed the word from “Figure 3” to “Figure 4” in text line 79 in the previous manuscript.

Point 4:

Fig. 6 and Fig. 7 should be shown with the horizontal axis as input RF power and the vertical axis as output RF power. The reader is not interested in the variable attenuator step. The text should also be revised.

Response 4:

Thank you for the valuable suggestions provided by the reviewer. In this revised manuscript, we have updated Figures 6 and 7 according to the reviewer's recommendation. In addition, we have also added the information of power gain in these two figures.

Point 5:

References should essentially be to peer-reviewed journals. If the reference is to a PhD thesis or Master's thesis, in many cases the reader will not be able to find the reference. (If a DOI is given, it should be stated).

Response 5:

In this revised manuscript, have included citations which are the published book, conference proceedings and theses that are available online.

Round 2

Reviewer 1 Report

Comments and Suggestions for Authors

After the corrections and additions introduced in the new version of the manuscript, I can suggest the manuscript, after some minor verification, for a journal publication in Particles.

Among the additions in the new version of the manuscript, I’m confused with the units in the right-hand side of Eq. (1) at page 3. To my understanding, in the right-hand side of the equation there is a product of charge (if e stands here for the electron charge) by RF voltage, which provides an energy. This energy is multiplied by unitless quality factor Q but divided by RF frequency, so the final units in my understanding are energy multiplied by time, in comparison with energy at the left-hand side of the equation. I’d suggest checking this before publication.

Another issue which could be easily improved is to provide direct hyper-references on the cited papers published at IPAC’22 conference.

Author Response

Response to Reviewer 1 Comments

After the corrections and additions introduced in the new version of the manuscript, I can suggest the manuscript, after some minor verification, for a journal publication in Particles.

Point 1:

Among the additions in the new version of the manuscript, I’m confused with the units in the right-hand side of Eq. (1) at page 3. To my understanding, in the right-hand side of the equation there is a product of charge (if e stands here for the electron charge) by RF voltage, which provides an energy. This energy is multiplied by unitless quality factor Q but divided by RF frequency, so the final units in my understanding are energy multiplied by time, in comparison with energy at the left-hand side of the equation. I’d suggest checking this before publication.

Response 1:

Thank you very much for addressing this point. It was our confusion about the definition of the quality factor, which is the stored energy from the RF field in the RF cavity and the power dissipated in the cavity wall per RF cycle. Thus, the energy gain of electron per RF cycle is the product of electron charge, accelerating voltage and quality factor as written in Equation (1) of this revised version.

Point 2:

Another issue which could be easily improved is to provide direct hyper-references on the cited papers published at IPAC’22 conference.

Response 2:

Thank you for your suggestion. In this revised manuscript, we have updated the citations to provide direct hyper-references to the papers published at IPAC’22 conference website. 
